# Effects of Compound Feed Attractants on Growth Performance, Feed Utilization, Intestinal Histology, Protein Synthesis, and Immune Response of White Shrimp (*Litopenaeus Vannamei*)

**DOI:** 10.3390/ani12192550

**Published:** 2022-09-23

**Authors:** Guilun He, Xin Chen, Qingtian Zeng, Wenbo Zhu, Zhengbang Chen, Beiping Tan, Shiwei Xie

**Affiliations:** 1Laboratory of Aquatic Animal Nutrition and Feed, School of Aquaculture, Guangdong Ocean University, Zhanjiang 524088, China; 2Grobest Group Holdings Limited (CN), Fuzhou 350000, China; 3Aquatic Animals Precision Nutrition and High-Efficiency Feed Engineering Research Centre of Guangdong Province, Zhanjiang 524088, China; 4Key Laboratory of Aquatic, Livestock and Poultry Feed Science and Technology in South China, Ministry of Agriculture, Zhanjiang 524088, China; 5Guangdong Provincial Key Lab of Pathogenic Biology and Epidemiology for Aquatic Economic Animals, Zhanjiang 524088, China

**Keywords:** *L. vannamei*, growth performance, feed utilization, feed attractants, protein synthesis

## Abstract

**Simple Summary:**

There have been few studies on the effects of compound feed attractants on the growth mechanism and health of aquatic animals. This study aimed to investigate the effects of compound feed attractants on the growth performance, feed utilization, intestinal histology, protein synthesis, and immune response of *Litopenaeus vannamei*. Seven diets were formulated, and shrimp (0.71 ± 0.00) g were distributed to seven groups of four replicates and fed for 7 weeks. In this study, diet supplemented with squid visceral powder, fish soluble, squid paste, and shrimp paste, and a diet supplemented with yeast extract, squid visceral powder, squid paste, and shrimp paste promoted the feed intake and growth performance and enhanced the antioxidative ability of *L. vannamei*. The combination of squid visceral powder, fish soluble, squid paste, and shrimp paste not only increased the protein synthesis, but also promoted the hepatopancreas and intestinal response of shrimp. Conversely, the diet supplemented with yeast extract, squid visceral powder, fish soluble, and shrimp paste may have some negative effects on the hepatopancreas and intestine of *L*. *vannamei*.

**Abstract:**

To investigate the effects of compound attractants on the growth performance, feed utilization, intestinal morphology, protein synthesis, and immune response of *L**itopenaeus vannamei*, the following seven diets were formulated: a positive control (P), a negative control (N), and five diets with compound attractants which were labeled as A, B, C, D, and E, each with four of five tested attractants (yeast extract, squid visceral powder, fish soluble, and squid paste, shrimp paste), respectively. Shrimp (0.71 ± 0.00 g) were distributed to seven groups of four replicates and fed for 7 weeks. Results showed that the final body weight, feed intake, specific growth rate, and weight gain of shrimp in the B and D groups were the greatest. Hemolymph total antioxidant capacity of shrimp in the B, D, and E groups reached the highest level. In the hepatopancreas, the activity of total nitric oxide synthase, malondialdehyde content, the expression levels of *sod*, *myd88*, *eif4e2*, and *raptor* in shrimp fed the B diet were the highest, and the highest levels of *dorsal* and *relish* were observed in the C group. In the intestine, intestinal muscle thickness and expression levels of *toll* and *eif2α* in the C group were the highest, while the highest expression level of *sod* and *relish* occurred in the B group. In summary, the B and E diets promoted the feed intake, growth performance and the antioxidant enzyme activity of *L. vannamei*. The C diet enhanced the protein synthesis of shrimp. Regression analysis indicated that the WG and FI of shrimp were increased as the dietary inclusion levels of squid paste and shrimp paste increased, while they were decreased as the dietary inclusion levels of yeast extract and fish soluble increased.

## 1. Introduction

*Litopenaeus vannamei* is one of the most widely cultured shrimp species worldwide due to its good taste, rapid growth, and excellent salinity adaptability [1]. In 2018, the global production of Pacific white shrimp (*Litopenaeus vannamei*) exceeded 4.9 million tons [2]. Fishmeal is one of the most important sources of protein in shrimp feed but, in recent years, the supply of fishmeal has struggled to satisfy the requirements of aquaculture [3]. New protein sources, especially plant protein sources, such as soybean meal, have been developed for fish meal alternatives [4,5,6]. However, the replacement of fish meal by plant protein is likely to reduce the palatability of feed intakes and the growth of rainbow trout (*Oncorhynchus mykiss*) [7], and white shrimp [8]. In addition, the feed intake process of *L. vannamei* is relatively slow compared with other aquatic animals, which results in a low feed utilization rate, as well as water pollution [9]. Thus, the attractants in the feed are critical to the growth and feed utilization of shrimp.

The feeding behavior of aquatic animals is closely related to species, environment, and feeding stimulation [10]. Feed attractants are divided into two types. One type is obtained from natural material, and the other type is produced through chemical synthesis [10]. A diet with good attractants can enhance the growth performance and feed utilization of aquatic animals [11,12]. A combination of any four of the five attractants were evaluated in this study. Yeast extracts are obtained from fermented yeast, which is rich in small peptides, oligosaccharides, and amino acids [13]. Yeast extract is more widely studied in shrimp than the other four attractants. The feed intake and feed utilization rate of *L. vannamei* are successfully increased by both 1% yeast hydrolysate and 1% brewer’s yeast [14], and the optimal feed intake and feed utilization rate of *L. vannamei* could be achieved when the yeast extract content was 2% [13]. In addition, some feed attractants are rich in nutrients, such as nucleotides, small peptides, and amino acids, which not only improve the growth rate but also enhance the immunity of aquatic animals. In previous studies, yeast extract has been shown to promote the immunity of *L. vannamei* [15], and increase the diversity of intestinal flora in *L. vannamei* [16]. Squid paste, squid visceral powder, fish soluble, and shrimp paste have a strong aromatic fishy odor. In addition, squid paste and squid visceral powder are obtained from squid, which is rich in amino acids, and squid extract promoted feeding and growth of Atlantic salmon (*Salmo salar* L.) in previous research [17], and squid viscera paste enhanced the feed intake and growth of *L. vannamei* [18]. Fish soluble is produced from herring and other oily fish during the manufacture of fish meal, which usually contains about 30% crude protein [19]. A previous study showed that dietary fish soluble supplementation promoted the growth of juvenile black sea bream (*Acanthopagrus schlegelii*) (Irm et al., 2020). Shrimp paste is historically made by naturally fermenting entire shrimp in a salt solution (25–30% salt by weight) in an ambient environment. Despite having a long shelf life of 6 months at ambient temperature and being immune to microbial contamination, traditional shrimp paste is mostly used as a condiment because of its high salt content [20]. A previous study reported that shrimp paste was beneficial to the growth performance of hybrid snakehead (*Channa maculata* ♀ × *Channa argus* ♂) (Fang et al., 2019).

Although there are many studies reporting the effects of attractants on the growth and feed utilization of shrimp, single attractants were used in most studies [3,12,21,22]. At the same time, a high content of a single attractant can bring negative effects to aquatic animals [23,24]. The combination of compound attractants in shrimp feed may overcome the disadvantage of single attractant in aquatic animals’ feed. As feed attractants, yeast extract, squid paste, squid viscera powder, shrimp paste, and fish soluble are widely used in aquatic animals. Until now, there have been few studies on the effects of compound feed attractants on the growth mechanism and health of aquatic animals. There has been no report on whether the combinations of squid extract, squid viscera powder, fish soluble, yeast extract, and shrimp extract can more effectively promote the growth, enhance the feed utilization efficiency, and improve the health of *L. vannamei*. This study aimed to assess dietary supplementation of five compound feed attractants on the growth performance, feed utilization, intestinal histology, protein synthesis, and immune response of *L. vannamei*.

## 2. Materials and Methods

### 2.1. Experimental Diets

Seven practical diets were formulated, namely a commercial feed (P) as a positive control, a negative control (N) without the addition of attractants, and five compound attractors, which were labeled as A, B, C, D, and E, each with four of five tested attractants (yeast extract, squid visceral powder, fish soluble, squid paste, and shrimp paste). The diets were prepared in the following steps. Firstly, the ingredients were ground and filtered through an 80-mesh sieve. Secondly, all ingredients were weighed and mixed to homogeneity, and then the oil mixture and water were added and mixed until homogeneous. Finally, the 1.0 and 1.5 mm diameter pellets were extruded using a pelletizer (Institute of Chemical Engineering, South China University of Technology, Guangdong, China) and dried in an electric oven at 65 ℃ for 30 min, before being stored frozen (−20 ℃) until use. Formulation and proximate composition analysis of the diets are given in Table 1.

### 2.2. Experimental Shrimp and Feeding

The juvenile Pacific white shrimp obtained from the nursery base of Guangdong Hengxing Seed Co., Ltd. (Zhanjiang, China) were fed a commercial feed to acclimate to experimental conditions for a week. A total of 840 healthy and uniform-sized shrimp [initial body weight of (0.71 ± 0.00) g] were distributed to 7 groups with 4 replicates (300 L fiberglass tank per replicate) and 30 shrimp in each tank. Shrimp were fed four times daily at 7:00, 12:00, 17:00, and 21:00 with 8–10% body weight per day for 7 weeks. In the first four weeks, shrimp were fed 1.0 mm diameter feed and, in the last three weeks, they were fed 1.5 mm of feed. During the experiment, 60% of the water was exchanged each day with seawater disinfected with chlorine dioxide to maintain water quality. Water temperature and salinity were measured every day in a range of 24–28 ℃ and 26–29‰.

### 2.3. Sampling and Chemical Analysis

At the end of the feeding trial, shrimp were fasted for 24 h then weighed and counted. Five shrimp were randomly selected from each replicate and stored in a freezer at −20 ℃ before analysis of whole body composition. Another five shrimp were used for analysis of muscle composition. Hemolymph from five shrimp was taken from the pericardial cavity using a 1-mL syringe, pooled and stored at 4 ℃ overnight, and then frozen centrifuged at 4000 rpm for 10 min. The supernatant was collected and stored at −80 ℃ until analysis of hemolymph biochemical parameters and enzymatic activities. Three shrimp were randomly selected from each tank, weighed, and the body length was measured to calculate the condition factor (CF). Then, the hepatopancreas was removed and weighed for calculating hepatopancreas somatic indices (HSI). About 50 mg of hepatopancreas and two intestines from two shrimp per tank were collected, placed in RNAlater, and placed at −80 ℃ before RNA extraction. Two hepatopancreas from each tank were stored at −80 ℃ before the analysis of enzyme activities. At last, intestines from two shrimp per tank were collected and fixed in 4% formaldehyde before the hematoxylin–eosin (H&E) stains.

The moisture, crude protein, crude lipid, and ash contents of the feed, muscles, and whole shrimp samples were determined according to the standard methods [25]. The crude protein content was determined by KjeltecTM 8400 (FOSS, Sweden); the crude lipid content was determined by an XT15 extractor (Ankom, USA), and the crude ash content was determined by a muffle and ashing at 550 ℃ for 6 h.

Total protein (TP) (A045-2-2), high density lipoprotein cholesterin (HDL-C) (A112-1-1), low density lipoprotein cholesterin (LDL-C) (A113-1-1), triglycerides (TG) (A110-1-1), total cholesterol (T-CHO) (A111-2-1), glutamic oxaloacetic transaminase (GOT) (C010-2-1), glutamic alanine transaminase (GPT) (C009-2-1), acid phosphatase (ACP) (A060-2-2), and alkaline phosphatase (AKP) (A059-2-2) activities in the hemolymph were measured. The malondialdehyde (MDA) (A003-1-2), total nitric oxide synthase (T-NOS) (A014-2-2), total antioxidant capacity (T-AOC) (A015-2-1), and total superoxide dismutase (T-SOD) (A001-1-2) in the hemolymph and hepatopancreas were also measured. All the above indexes were determined by kits (Nanjing Jiancheng Bioengineering Company, China), and the determination methods came from the instruction manual.

The midgut was placed in formaldehyde for 24 h. After dehydration in a graded series of ethanol, the sample is embedded in paraffin. Sections with a thickness of 5–7 μm were stained with H&E and observed through a Nikon ECLIPSE 80I microscope (Nikon, Japan). Ten microscope fields were randomly selected for each section sample for measurement and statistics. Intestinal fold height (FH), fold width (FW), and muscle thickness (MT) were measured.

Total RNA was extracted from the hepatopancreas and intestines using the TransZol Up Plus RNA kit (TransGen, China). Agarose gel electrophoresis and Nanodrop 2000, (Thermo Fisher Scientific, USA) were used to assess RNA quality and concentration. cDNA synthesis was performed using the PrimeScript™ RT–PCR Kit according to the manufacturer’s (Takara, Japan) instructions. Real-time PCR was performed by quantitative thermal cycler (LightCycler 480 Roche Diagnostics, Switzerland) in a reaction volume of 10 μL, which contained 5 μL of 2 × SYBR^®^ Green Pro Taq HS Premix II (Accurate Biotechnology (Hunan) Co., Ltd. Accurate Biotechnology, China), 1 μL of cDNA, 0.4 μL of forward and reverse specific primers, and 3.2 μL of nuclease-free water, with a denaturation step at 95 ℃ for 2 min; 95 ℃ for 15 s, 60 ℃ for 15 s, and 72 ℃ for 20 s, for 40 cycles. Based on our preliminary experimental results, the gene elongation factor 1α (*ef-1α*) was used as the reference gene. The relative gene expression was analyzed by the 2^−ΔΔCT^ method [26]. The expression level of the normalized gene in the *p* group was set as 1. All primers were designed using the PrimerQuest tool (National Center for Biotechnology Information, USA), and the primer sequences are shown in Table 2.

### 2.4. Calculation Formula and Statistical Analysis

The following equations were employed:

Survival rate (SR, %) = 100 × final shrimp number/initial shrimp number;

Weight gain rate (WGR, %) = 100 × [final mean weight (g) − initial mean weight (g)]/initial weight (g);

Feed conversion ratio (FCR) = Total dry feed intake (g)/[final weight (g) − initial weight (g)];

Specific growth rate (SGR, %) = 100 × [Ln final mean weight (g) − Ln initial mean weight (g)/feeding days (D)];

Feed intake (FI, g) = Feed consumed (g)/Number of shrimp;

Hepatosomatic index (HSI, %) = Body weight (g)/body length (cm)^3^ × 100;

Condition factor (CF, %) = Weight of hepatopancreas/body weight × 100;

The results in this experiment were presented as mean ± standard error (SEM). Statistically significant differences were established by using one-way analysis of variance (ANOVA) at a 5% level of probability, and differences between means were compared using the Duncan range test. Regression analysis was performed to investigate the effects of yeast extract, squid visceral powder, fish soluble, squid paste, and shrimp paste on the WGR and FI. Attractants in each diet (A–E) were used as the independent variable, while WG and FI were used as the dependent variables. Statistical analysis was carried out using SPSS 21 (SPSS Inc., Chicago, IL, USA).

## 3. Results

### 3.1. Growth Performance, Feed Utilization, and Morphological Parameters

The growth performance, feed utilization, and morphological parameters of juvenile *L. vannamei* fed experimental diets are shown in Table 3. The FBW, WGR, and SGR of shrimp in the B and D groups were significantly higher than those in the *p* group (*p* < 0.05). The SR, FCR, and HIS of shrimp among all the experimental groups were found to have no significant differences (*p* > 0.05). The FI of shrimp fed the B, C, and D diets were significantly higher than those fed the *p* diet (*p* < 0.05). The CF of shrimp in the N group was significantly higher than that in other groups (*p* < 0.05). The effects of five feed attractants on the WGR of white shrimp were determined using multiple regression analysis, and the regression equations were determined as follows: WGR = 1123.78 − 12.71 × yeast extract level − 13.10 × fish soluble level + 26.58 × squid paste level + 15.21 × shrimp paste level, R^2^ = 1.000; FI= 10.360 − 0.060 × yeast extract level − 0.055 × fish soluble level + 0.110 × squid paste level + 0.195 × shrimp paste level, R^2^ = 1.000. Here, R^2^ = 1.000 indicated a high correlation between the regression equations and the model. In the equations about WG and FI, the coefficients of squid paste and shrimp paste are positive, and the coefficients of yeast extract and fish soluble are negative, which indicates that the WG and FI of shrimp were increased as the dietary inclusion levels of squid paste and shrimp paste increased, while they were decreased as the dietary inclusion levels of yeast extract and fish soluble increased.

### 3.2. Whole Body and Muscle Composition of the Shrimp L. vannamei

The whole body and muscle composition of juvenile *L*. *vannamei* fed experimental diets are shown in Table 4. In the whole body of shrimp, the moisture of shrimp in the E group was significantly higher than that in the N, C, and D groups (*p* < 0.05); ash showed no differences among all the experimental groups (*p* > 0.05); the crude lipid of shrimp in B group was significantly higher than that in other groups (*p* < 0.05); the crude protein of shrimp in the N, B, and C groups was significantly higher than in the E group (*p* < 0.05). In the muscle, moisture, ash, crude lipid, and crude protein of the muscle showed no differences among the all experimental groups (*p* > 0.05).

### 3.3. Hemolymph Biochemical Parameters

Hemolymph biochemical parameters are presented in Figure 1. The LDL-C of shrimp in the N group was significantly higher than that in other groups (*p* < 0.05), and LDL-C in the D group was significantly lower than that in other groups (*p* < 0.05). Shrimp in the C and E groups had significantly higher HDL-C content than those in the P, N, A, B, and D groups (*p* < 0.05), and shrimp in the P group had significantly higher HDL-C content than those in the A and B groups (*p* < 0.05). Those in the N group had significantly higher HDL-C content than those in the B group (*p* < 0.05). Here, T-CHO, TP, TG, and MDA contents were similar among all the experimental groups (*p* > 0.05).

### 3.4. Evaluation of Hemolymph Enzyme Activities

Hemolymph enzyme activities are presented in Figure 2. The GPT activity of shrimp fed the E diet was significantly higher than that of those fed the B diet (*p* < 0.05). The T-AOC of shrimp fed the B, D, and E diets was significantly higher than those fed the N diet (*p* < 0.05). The GOT, ACP, AKP, T-NOS, and T-SOD activities of shrimp showed no differences among all the experimental groups (*p* > 0.05).

### 3.5. Enzyme Activities in the Hepatopancreas

Hepatopancreatic enzyme activities are presented in Figure 3. The T-NOS activity of shrimp in the C group was significantly higher than in the P, B, C, and E groups (*p* < 0.05), and the N group was significantly higher than that in the B, and E groups (*p* < 0.05). All the T-SOD activities showed no differences among all the experimental groups (*p* > 0.05). The MDA content of shrimp in the B group was significantly higher than that in the P, A, C, D, and E groups (*p* < 0.05). The T-AOC of shrimp in the B group was significantly higher than that in the E group (*p* < 0.05).

### 3.6. Intestinal Histology

The intestinal histology of shrimp is presented in Figure 4. The intestine tissues of shrimp fed the compound feed attractants showed that intestinal epithelium connected closely, and it could be observed that the intestinal fold of shrimp in the B and D groups were significantly different from those in the P group (*p* < 0.05). The FH and FW of shrimp showed no differences among all the experimental groups (*p* > 0.05). The MT of shrimp fed the C diet was thicker than those fed the E diet (*p* < 0.05) (Table 5).

### 3.7. Immune Response and Protein Synthesis Related Genes Expressions in Hepatopancreas

Immune response and protein synthesis related gene expression in the hepatopancreas of *L. vannamei* are presented in Figure 5 and Figure 6, respectively. The expression level of *toll* in shrimp fed the N diet was significantly higher than those fed the C and D diets (*p* < 0.05). The expression level of *sod* in shrimp fed the B, D, and E diets was significantly higher than those fed the P, N, and A diets (*p* < 0.05), and the levels of *sod* in shrimp fed the C diet was significantly higher than those fed the N diet (*p* < 0.05). The expression level of *myd88* in shrimp fed the B diet was significantly higher than those fed the N, A, D, and E diets (*p* < 0.05), and in shrimp fed the *p* diet it was significantly higher than those fed the D diet (*p* < 0.05). The expression level of *dorsal* in shrimp fed the C diet was significantly higher than those fed the other diets (*p* < 0.05), and in shrimp fed the A and E diets it was significantly higher than those fed the B and D diets (*p* < 0.05). The expression level of *relish* in shrimp fed the C diet was significantly higher than those fed the other diets (*p* < 0.05), and in shrimp fed the P and A diets it was significantly higher than those fed the N, B, D, and E diets (*p* < 0.05) (Figure 5). The expression level of *eif4a* in shrimp fed the D and E diets was significantly lower than those fed the other diets (*p* < 0.05). The expression level of *eif4e2* in shrimp fed the A and B diets was significantly higher than those fed the C, D, and E diets (*p* < 0.05), and in shrimp fed the P and N groups it was significantly lower than those fed the A diet (*p* < 0.05). The expression level of *eif2α* in shrimp fed the N diet was significantly higher than those fed the P, B, C, D, and E diets (*p* < 0.05), while in shrimp fed the A diet it was significantly higher than those fed the D diet (*p* < 0.05). The expression level of *raptor* in shrimp fed the B diet was significantly higher than those fed the other diets (*p* < 0.05), while in shrimp fed the P diet it was significantly higher than those fed the A, C, D, and E diets (*p* < 0.05); additionally, in shrimp fed the N and A diets, it was significantly higher than those fed the C, D, and E diets (*p* < 0.05). The expression level of *tor* in shrimp fed the N and E diets was significantly higher than that of those fed the P, A, B, and C diets (*p* < 0.05), while in shrimp fed the D diet it was significantly higher than those fed the B and C diets (*p* < 0.05). The expression level of *s6k* in shrimp showed no differences among all experimental groups (*p* > 0.05) (Figure 6).

### 3.8. Intestinal Genes Expression

Intestinal gene expressions are presented in Figure 7. The expression level of *toll* in shrimp fed the C diet was significantly higher than those fed the A and E diets (*p* < 0.05). The expression level of *sod* in shrimp fed the B diet was significantly higher than those fed the P, N, A, D, and E diets (*p* < 0.05), while in shrimp fed the C diet it was significantly higher than those fed the E diet (*p* < 0.05). In shrimp fed the P, N, A, and C diets, *dorsal* expression was significantly higher than those fed the B, D, and E diets (*p* < 0.05). The expression level of *relish* in shrimp fed the B diet was significantly higher than those fed the other diets (*p* < 0.05), while in shrimp fed the C diet it was significantly higher than those fed the P, N, A, D, and E diets (*p* < 0.05). The expression level of *tor* in shrimp fed the N diet was significantly higher than those fed the E diet (*p* < 0.05). The expression level of *s6k* in shrimp fed the P diet was significantly lower than those fed the other diets (*p* < 0.05).

## 4. Discussion

Few reports have explored the suitable compound attractants for *L. vannamei*. In this study, based on growth performance, the shrimp fed the B and D diets obtained better growth performance than those fed the other diets. Regression analysis showed that squid paste and shrimp paste contributed more to the WG and FI, while yeast extract and fish soluble may have negative effects. Many previous studies also revealed that squid paste was an excellent attractant in aquafeed. A previous study showed that the *L. vannamei* fed a diet containing 5% squid paste gained greater growth performance than those fed the diet without squid paste [11]. It was reported that squid extract was expected to enhance the palatability and feeding rate of gibel carp (*Carassius auratus gibelio*) fed diets with or without meat and bone meal [22]. Salmon (*Salmo salar* L.) performs better during the parr-smolt transformation period when fed 5 g kg^−1^ squid extract [17]. Replacing 60% FM protein with soybean protein concentrate (SPC) and a mixture (total 15%) of fish soluble (FS), krill meal (KM), and squid meal (SM) each at 5%, respectively, provided red sea bream (*Pagrus major*) enough amino acids to balance the nutrition, and those would have acted as feeding attractants [27]. In accordance with our results, dietary 3% shrimp paste supplementation may improve the growth performance of hybrid snakehead by enhancing FI [28]. However, a previous study showed that feeding Chinese perch (*Siniperca chuatsi*) with 5% squid paste had no effects on growth performance and feed utilization [29]. Contrary to our findings, there were previous studies which showed that the feed conversion ratio of white shrimp was reduced by consuming a yeast culture feed supplement containing 1.0 or 1.5 g kg^−1^ [30]. In addition, a previous study showed that the three most effective attractants were herring processing by-products, sand eel hydrolysate, and hydrolysate by-products from the shrimp industry, which are more effective than squid in the diet of Atlantic cod (*Gadus morhua*) [31]. The different results may be caused by the differences in species and the inclusion levels of attractants. It has been demonstrated that crustaceans can become desensitized to certain chemical stimuli if subjected to inappropriately high concentrations [10]. Dietary 60 g kg^−1^ flash dried yeast (FDY) supplementation significantly reduced growth, feed utilization, and protein retention of white shrimp, while high levels supplementation (≥ 60 g kg^−1^) of FDY had negative effects on the growth response [23]. In our experiment, the shrimp in the B and D groups appeared to demonstrate the best growth performance and FI, suggesting that the compound attractants used in the B and D diets were more suitable for the growth of white shrimp.

Hemolymph biochemical parameters usually reflect the metabolism and health status of aquatic animals [32,33]. Additionally, HDL and LDL are essential lipid droplets surrounded by specific proteins, their main function being to transport cholesterol [34]. Indeed, HDL plays a key role in reverse cholesterol transport, which transports cholesterol to the liver, while LDL is the main cholesterol carrier in the circulation, and its physiological function is to carry cholesterol to cells [35,36]. The reduction in HDL-C and LDL-C will lead to dysfunction of energy storage, and lipid transportation [34,37]. In our study, the LDL-C content of hemolymph in shrimp fed the N and A diets were highest, which was significantly higher than those fed the D diet. Thus, it was inferred that the cellular ability to remove cholesterol from bile in shrimp fed the N and A diets may be the strongest, while in shrimp fed the D diet may be the weakest. The HDL-C content of hemolymph in shrimp fed the C and E diets was the highest, in shrimp fed the P diet it was significantly higher than those fed the A and B diets, while in shrimp fed the B diet it was lower than those fed the N diet with the lowest content. Thus, it was inferred that the shrimp in the C and E groups may have the strongest ability to transport cholesterol to the liver, while the B group may be the weakest.

The increase in GPT and GOT in hemolymph are associated with tissue destruction, cell necrosis in the hepatopancreas and other tissues [38,39]. In this study, the GPT activity of shrimp in the E group was significantly higher than that in the B group, which indicated the shrimp fed the E diet might suffer a greater degree of hepatopancreas damage.

The enzyme SOD is the main antioxidant enzyme in the organism, which generates hydrogen peroxide (H_2_O_2_) after the conversion of superoxide anion (O^2−^) by SOD [40,41]. In our study, the T-SOD activity level in the hepatopancreas of shrimp fed the A diet was the highest, and those fed the B, and E diets showed the lowest activity. The *sod* expression levels of the hepatopancreas in shrimp fed the B, D, and E diets were upregulated compared to the control groups, while the shrimp fed the B and C diets favored the upregulation of *sod* expression levels in the intestine. The MDA is a lipid peroxidation decomposition product of polyunsaturated fatty acids (PUFAs), which has a cytotoxic effect [42]. In this study, the highest MDA levels and increased damage to the hepatopancreas were observed in the B group, but in the hemolymph there was no significant difference among all experimental groups. According to the levels of T-SOD, *sod*, and MDA in hepatopancreas and the expression level of *sod* intestine, it was inferred that shrimp in the A group obtained good antioxidant capacity, and that vigorous lipid peroxidation decomposition metabolic activity happened in shrimp in B group. The reason for this phenomenon might be that the B diet contains more aquatic animal fat sources. However, the trend of T-SOD and the expression levels of *sod* in the hepatopancreas of shrimp in the B and D groups were opposite, which may be caused by the lower translation level compared to the transcription level. Furthermore, T-AOC is related to the antioxidant function of the organism and reflects the status of the non-enzymatic system and the antioxidant enzyme system in the organism [43]. In this study, the T-AOC of hemolymph in the B, D, and E groups was significantly enhanced compared to the N group, while the highest T-AOC of hepatopancreas occurred at the highest levels in the A group, indicating that A, B, D, and E diets may strengthen the non-enzymatic system and the antioxidant enzyme system in the organism of shrimp to enhance the immunity of the *L. vannamei*. Here, NOS is the key enzyme for NO synthesis, and nitric oxide (NO) plays an important role in the innate immune systems of many species for antimicrobial defense, when bacteria or pathogens attack the organism to produce nitrous oxide (NO) [44,45,46]. In the present study, the highest level of T-NOS activity in the hepatopancreas was found in the A group, while there was no significant difference in hemolymph among all groups, which implied that the A diet was beneficial to enhance disease resistance in the hepatopancreas of *L. vannamei*. The highest levels of T-NOS and T-AOC of shrimp were found in the A group, which was beneficial to enhance disease resistance in the hepatopancreas of *L. vannamei*.

The intestine serves as a critical organ for nutrient uptake and utilization as well as a crucial pathogen defense [47]. The function of the intestinal wall depends on the length of the fold [48]. The muscularis is related to intestinal motility and its thickness can show the ability of intestinal peristalsis [49]. Muscle thickness plays a role in intestinal digestion and absorption, thus, increased muscle thickness may enhance the intestinal digestion and absorption ability of shrimp [50]. In our study, it could be observed that the intestinal folds of shrimp in the B and D groups were significantly different from that in the P group, and that the B and D diets alleviated the swelling of the intestinal fold of shrimp, which was beneficial to restore the histology of intestinal fold. In addition, there were no significant differences in intestinal fold height and width, while the intestinal muscle thickness of the C group was highest (Table 5), which indicated that the shrimp fed the C diet may have a higher ability of intestinal peristalsis. Previous studies reported that shrimp fed the diet containing 30 g/kg nucleotide-rich yeast had higher intestinal fold height, fold width, and enterocyte height than those fed the control diet, and shrimp fed the 50 g/kg nucleotide-rich yeast diet had the highest microvillus height among all treatments [51]. The results of the present study were similar to a previous study for other aquatic animals. A previous study found that hybrid snakehead (*Channa*
*maculata* ♀ × *Channa argus* ♂) fed diets supplemented with shrimp paste had no influence on villi length and muscle thickness, but increased the number of goblet cells, which was why it was suggested that feed attractant may have a protective effect on the intestines of snakeheads [28].

*Toll* is an important innate immune receptor, which is achieved by binding the adapter complex to induce the NF-κB signaling pathway [52,53]. The *myd88*, *dorsal*, and *relish* are the major components of the NF-κB signaling pathway and are involved in the regulation of many inflammatory responses [48,54,55]. Previous studies have shown that *Vibrio anguillarium* greatly enhanced the expression levels of relish and dorsal, and it was discovered that silencing *dorsal* significantly decreased the expression of the antimicrobial peptide penaeidin 5 [56,57]. In the present study, in the hepatopancreas, the expression level of *toll* in shrimp fed the C, and D diets was downgraded compared to those fed the N diet. The highest expression level of *myd88* in shrimp was observed in the B group, while the highest expression levels of *dorsal* and *relish* occurred in the C group, and those in the D group showed the lowest expression levels of *myd88*, *dorsal*, and *relish*. In contrast, in the intestine, the expression level of *toll* in shrimp fed the C diet was significantly higher than those fed the A and E diets, and the expression level of *dorsal* in shrimp fed the P, N, A, and C diets was significantly higher compared to those fed the B, D, and E diets. The *relish* in shrimp fed the B diet obtained the highest expression level, followed by those fed the C diet. From our results, we can learn that the expression levels of immune related genes in shrimp fed the B and C diets were upregulated, and levels of those fed the E diet were downgraded. Therefore, our results indicated that the shrimp fed the B and C diets showed improved antioxidant capacity and immune response, which is consistent with the results of hepatopancreas enzyme activity, and that the immunity of the shrimp fed the E diet seems impaired.

The protein content is an important nutritional indicator of shrimp products. Some previous research indicated that the feed attractant does not exert a significant effect on the proximate composition of shrimp [58,59]. In the present study, proximate compositions in whole body and muscle were not significantly influenced by dietary compound feed attractant supplementation except for crude protein content in whole body compound feed attractants, which in the N, B, and C groups was significantly higher than that in the E group.

The main mechanisms governed by the rapamycin signaling pathway include promotion of protein synthesis and suppression of autophagy [60], and a significant modulator of nutrient-sensing and metabolism is *tor* signaling [61]. Indeed, *tor* controls several elements involved in the initiation and extension phases of translation to govern the synthesis of proteins [62], and controls the eukaryotic translation initiation factor 4e binding protein (*4ebp*) and causes the downstream target ribosomal protein S6 synthesis protein kinase (*s6k*) to bind to *raptor* [3]. The 40 S ribosomal subunit’s capped 5′ ends are attracted by the eIF3-eIF4G-eIF4E (eif4a) interaction, which creates many contact points for other initiation factors. This is the first step in the initiation of translation [40,63]. In our study, the expression level of *eif4a* in hepatopancreas of shrimp fed the D and E diets was downgraded compared to shrimp fed the other diets, which indicated that protein synthesis may be negatively affected by being fed the D and E diets. The expression levels of *eif4e2* in the hepatopancreas of shrimp fed the A and B diets were upregulated, meaning that they facilitated protein synthesis. The expression level of *eif2α* in the hepatopancreas in the negative control was the highest with high protein synthesis activity, which was downgraded to the lowest level in shrimp fed the D diet, leading to unfavorable protein synthesis. In contrast, the expression level of *raptor* in shrimp fed the B diet was the highest, while the expression levels in shrimp fed the C, D, and E diets were significantly downgraded. The shrimp in the N group appeared to have the highest level of *tor* gene expression, which was downgraded from those in the A, B, and C groups and, surprisingly, the highest expression level of *tor* in the hepatopancreas was also observed in shrimp fed the E diet. Unlike in the hepatopancreas, in the intestine, the expression level of *tor* in shrimp fed the N diet was significantly higher than those fed the E diet, and the lowest expression level of *s6k* in shrimp appeared in the P group. In conclusion, the crude protein content of body composition was closely associated with the expression of protein synthesis related genes. The highest crude protein content of body composition was found in the N, B, and C groups, and the lowest crude protein content occurred in the E group, which is generally consistent with protein synthesis related genes expression results.

## 5. Conclusions

In this study, the diet supplemented with squid visceral powder, fish soluble, squid paste, and shrimp paste (B), and a diet supplemented with yeast extract, squid visceral powder, squid paste, and shrimp paste (D) promoted the feed intake and growth performance and enhanced the antioxidative ability of *L. vannamei*. The combination of squid visceral powder, fish soluble, squid paste, and shrimp paste (B) not only increased the protein synthesis, but also promoted the hepatopancreas and intestinal response of shrimp. Regression analysis indicated that the WG and FI of shrimp were increased as the dietary inclusion levels of squid paste and shrimp paste increased, while they were decreased as the dietary inclusion levels of yeast extract and fish soluble increased.

## Figures and Tables

**Figure 1 animals-12-02550-f001:**
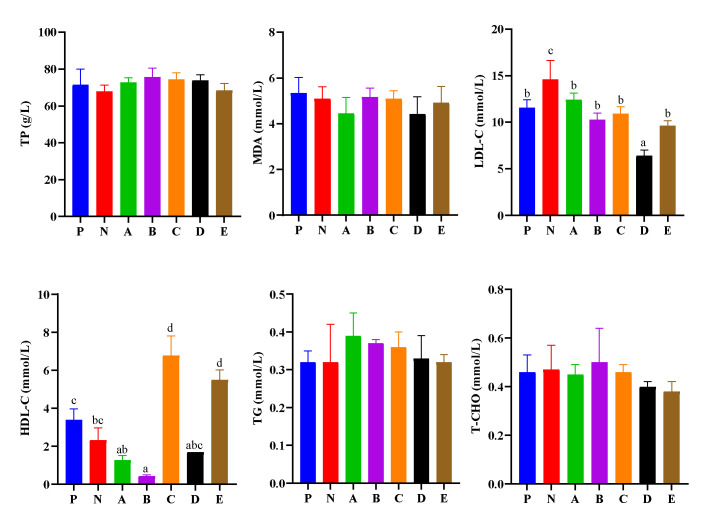
Effects of compound feed attractants on hemolymph biochemical parameters of white shrimp. Abbreviations are as follows: TP, total protein; MDA, malondialdehyde; LDL-C, low density lipoprotein cholesterin; HDL-C, high density lipoprotein cholesterin; TG, triglyceride; T-CHO, total cholesterol. Vertical bars represent the mean ± SEM (n = 4). Mean values with different letters are significantly different (*p* < 0.05).

**Figure 2 animals-12-02550-f002:**
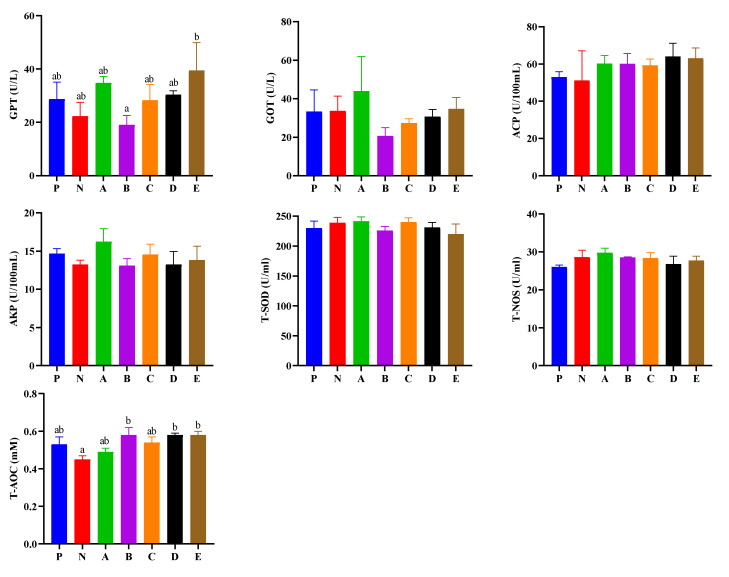
Effects of compound feed attractants on hemolymph enzyme activities of white shrimp. Abbreviations are as follows: GPT, glutamic pyruvic transaminase; GOT, glutathione transferase; ACP, acid phosphatase; AKP, alkaline phosphatase; T-SOD, total superoxide dismutase; T-NOS, total nitric oxide synthase; T-AOC, total antioxidant capacity. Vertical bars represent the mean ± SEM (n = 4). Mean values with different letters are significantly different (*p* < 0.05).

**Figure 3 animals-12-02550-f003:**
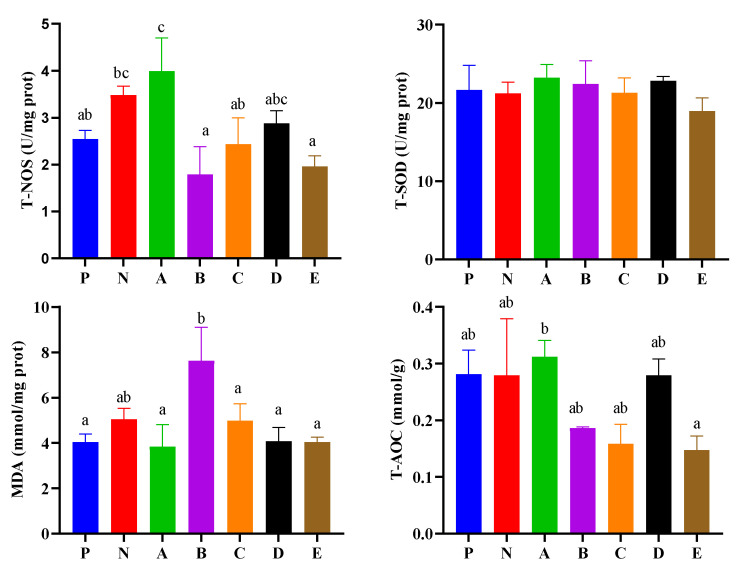
Effects of compound feed attractants on hepatopancreas enzyme activities of white shrimp. Abbreviations are as follows: T-NOS, total nitric oxide synthase; T-SOD, total superoxide dismutase; MDA, malondialdehyde; T-AOC, total antioxidant capacity. Vertical bars represent the mean ± SEM (n = 4). Mean values with different letters are significantly different (*p* < 0.05).

**Figure 4 animals-12-02550-f004:**
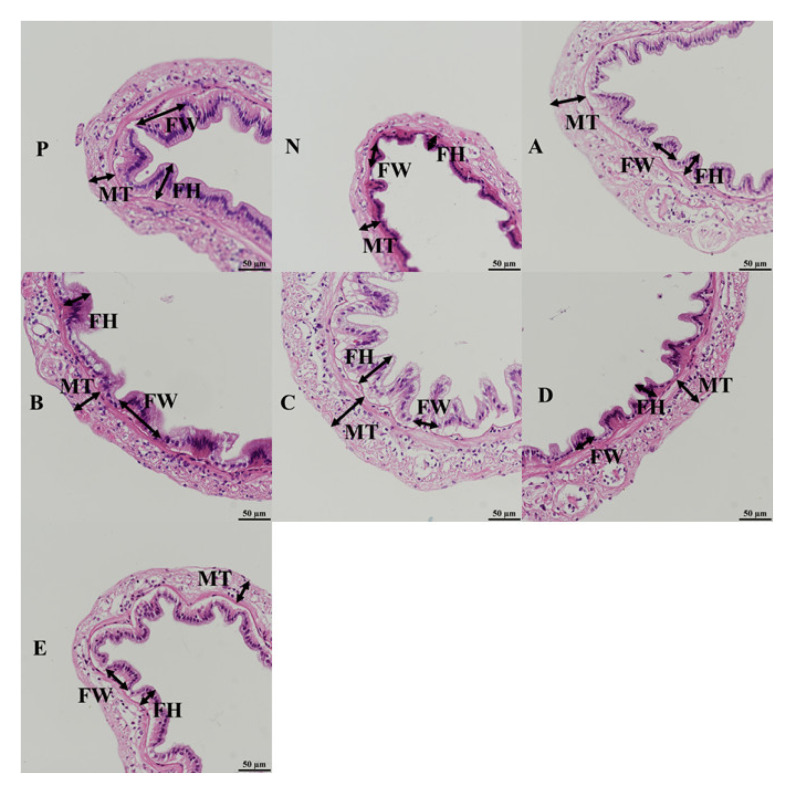
Photomicrographs (P, N, A, B, C, D, and E) of the midgut of white shrimp fed compound feed attractants diets. Arrows MT, FH, and FW represent intestinal muscle thickness, fold height, and fold width, respectively.

**Figure 5 animals-12-02550-f005:**
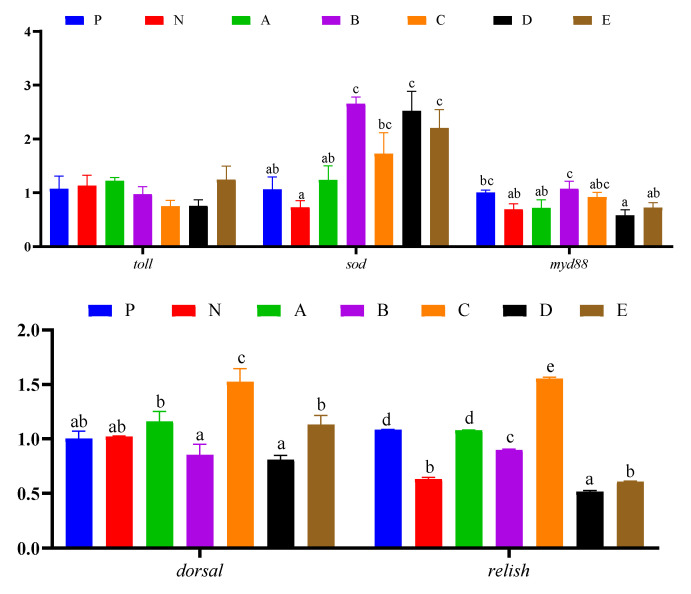
Effects of compound attractants on immunity synthesis related genes expression in the hepatopancreas of white shrimp. Upright bars denote the mean ± SEM (n = 4). Bars labeled with different letters denote significant differences (*p* < 0.05) among groups.

**Figure 6 animals-12-02550-f006:**
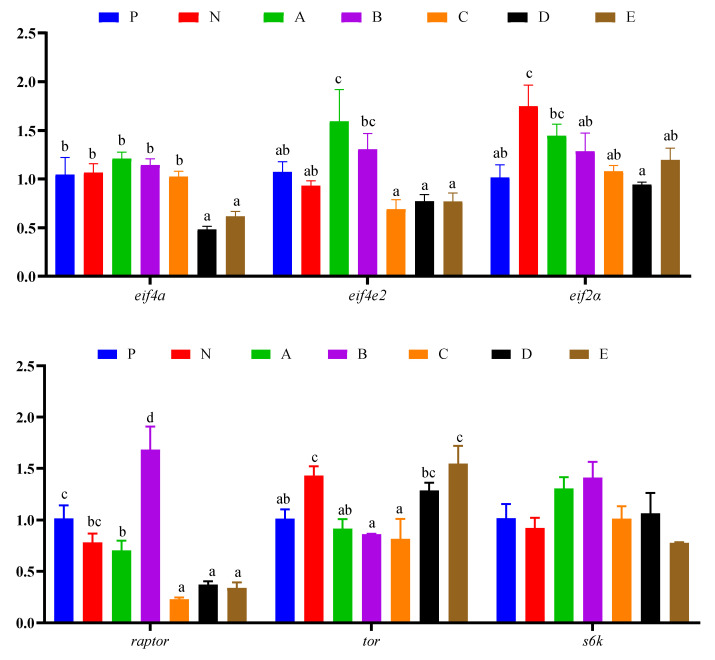
Effects of compound attractants on protein synthesis related gene expression in the hepatopancreas of white shrimp. Upright bars denote the mean ± SEM (n = 4). Bars labeled with different letters denote significant differences (*p* < 0.05) among groups.

**Figure 7 animals-12-02550-f007:**
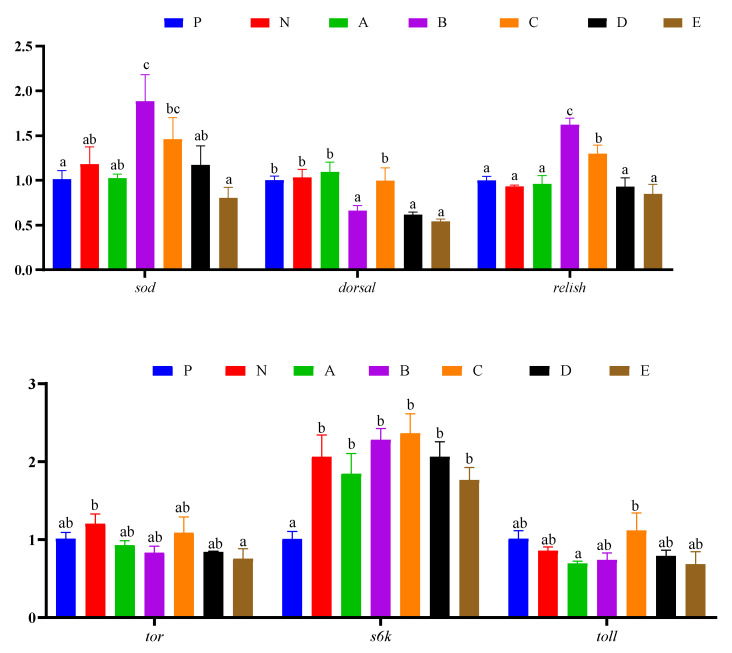
Effects of compound attractants on the intestinal gene expression levels of white shrimp. Upright bars denote the mean ± SEM (n = 4). Bars labeled with different letters denote significant differences (*p* < 0.05) among groups.

**Table 1 animals-12-02550-t001:** Formulation of supplement experimental diets (%) (wet basis).

Formula	P	N	A	B	C	D	E
Domestic fish meal	15.38	15.38	15.38	15.38	15.38	15.38	15.38
Brown fish meal	6.15	6.15	6.15	6.15	6.15	6.15	6.15
Chicken by-product meal	4.00	4.00	4.00	4.00	4.00	4.00	4.00
Peanut meal	6.00	6.00	6.00	6.00	6.00	6.00	6.00
Fish by-product meal	7.00	7.00	7.00	7.00	7.00	7.00	7.00
Shrimp meal	6.00	6.00	6.00	6.00	6.00	6.00	6.00
Corn protein powder	5.00	5.00	5.00	5.00	5.00	5.00	5.00
Cottonseed protein	4.00	4.00	4.00	4.00	4.00	4.00	4.00
Soybean meal	7.00	15.00	7.00	7.00	7.00	7.00	7.00
Wheat flour	17.20	17.20	17.20	17.20	17.20	17.20	17.20
Choline chloride	0.31	0.31	0.31	0.31	0.31	0.31	0.31
Monocalcium phosphate	1.23	1.23	1.23	1.23	1.23	1.23	1.23
Calcium chloride	0.98	0.98	0.98	0.98	0.98	0.98	0.98
Shell powder	0.62	0.62	0.62	0.62	0.62	0.62	0.62
Yeast extract	1.30	0	2.00	0	2.00	2.00	2.00
Squid visceral powder	4.80	0	2.00	2.00	0.00	2.00	2.00
Fish soluble	1.90	0	2.00	2.00	2.00	0	2.00
Squid paste	0	0	2.00	2.00	2.00	2.00	0
Shrimp paste	0	0	0	2.00	2.00	2.00	2.00
Soybean phospholipid	1.23	1.23	1.23	1.23	1.23	1.23	1.23
Lysine	0.37	0.37	0.37	0.37	0.37	0.37	0.37
Additive premix^1^	9.53	9.53	9.53	9.53	9.53	9.53	9.53
Proximate composition (% wet weight)							
Crude protein	43.00	42.07	44.64	45.57	44.37	43.32	45.11
Crude lipid	6.60	5.84	5.77	5.95	5.17	6.16	6.14
Moisture	6.78	6.44	6.44	6.88	7.35	6.48	8.19
Ash	16.13	15.97	16.41	16.30	16.36	15.56	16.13

Note: Additive premix^1^ (kg^−1^ of mixture) containing shrimp and crab with trace element premixed (F15) 24.60 g, fermented soybean meal (Yotoya) (F12) 46.20 g, Be-Yi-Po 9.80 g, compound premix–feeding kahao (shrimp) E 3.10 g, vitamin premix (E14) for shrimp and crab 4.90 g, le tong ning 1.80 g, mixed feed additive–amino acid preparation (feed tonic) (F11) 3.70 g, yuchang special eel multidimensional D17 1.20 g.

**Table 2 animals-12-02550-t002:** Primer sequence used for quantitative real-time PCR.

Primers	Sequences (5′-3′)	GenBank No.
*ef-1a-R*	GTATTGGAACAGTGCCCGTG	JF288785.1
*ef-1a-F*	ACCAGGGACAGCCTCAGTAAG	
*toll-R*	CCTCGCACATCCAGGACTTTTA	/
*toll-F*	GACCATCCCTTTTACACCAGACT	
*sod-R*	CAGAGCCTTTCACTCCAACG	/
*sod-F*	GCAATGAATGCCCTTCTACC	
*tor-R*	GGGTGTTTGTGGACGGA	MN398907.1
*tor-F*	TGCCAACGGGTGGTAGA	
*s6k-R*	CCGCCCTTGCCCAAAACCT	XM_027368997.1
*s6k-F*	GCAAGAGGAAGACGCCATA	
*myd88-R*	GGGAGTGGCAGAAACTTATC	/
*myd88-F*	GTGCACCAGAGTCATTGTAG	
*dorsal-R*	CGTAACTTGAGGGCATCTTC	FJ998202.1
*dorsal-F*	TGGGGAAGGAAGGATGG	
*relish-R*	GGCTGGCAAGTCGTTCTCG	EF432734
*relish-F*	CTACATTCTGCCCTTGACTCTGG	
*eif4a-R*	GTAGTCGCCAAGAGCAAGCAC	/
*eif4a-F*	CTCCAAATCCCAAGAAACCCA	
*eif4e2-R*	GTCCTCCTGGAAGCGTA	/
*eif4e2-F*	TGGAATCAAACCTATGTGGG	
*eif2a-R*	CTAATGCCCTAAGACCATCCTG	/
*eif2a-F*	GAATAAACCTAATCGCACCACC	
*raptor-R*	TCACAATCCAAGGTCCAG	XM_027360909.1
*raptor-F*	CTGCTTTCCAGGCTACTC	

Abbreviations are as follows: *F*, forward primer; *R*, reverse primer; E*f-1α*, elongation factor 1α; *dorsal*, transcription factor p65-like; *relish*, nuclear factor NF-kappa-B p105 subunit-like; *toll*, protein toll; *sod* superoxide dismutase; *tor*, target of rapamycin; *s6k*, ribosome protein S6 kinase; *myd88*, myeloid differentiation primary response gene 88; *eif4a*, eukaryotic translation initiation factor 4A; *eif4e2*, eukaryotic translation initiation factor 4E family member 2; e*if2a*, eukaryotic initiation factor 2α; *raptor*, regulatory-associated protein of TOR.

**Table 3 animals-12-02550-t003:** Effects of compound feed attractants on feed utilization and morphological parameters of the *Litopenaeus vannamei*.

Groups	FBW (g)	SR (%)	FI (g)	FCR	WGR (%)	SGR (%)	CF	HSI (%)
P	8.23 ± 0.14 ^a^	97.49 ± 0.83	10.05 ± 0.11 ^a^	1.33 ± 0.03	1066.04 ± 20.56 ^a^	4.90 ± 0.03 ^a^	0.64 ± 0.01 ^a^	4.46 ± 0.23
N	8.44 ± 0.34 ^ab^	93.33 ± 1.35	10.52 ± 0.16 ^ab^	1.36 ± 0.03	1094.04 ± 47.85 ^ab^	4.95 ± 0.08 ^ab^	0.71 ± 0.02 ^b^	3.97 ± 0.34
A	8.64 ± 0.11 ^ab^	94.16 ± 2.49	10.35 ± 0.24 ^ab^	1.30 ± 0.02	1125.3 ± 17.17 ^ab^	5.00 ± 0.02 ^ab^	0.61 ± 0.01 ^a^	4.26 ± 0.05
B	9.04 ± 0.23 ^b^	95.82 ± 0.83	10.86 ± 0.23 ^b^	1.30 ± 0.18	1181.16 ± 32.14 ^b^	5.09 ± 0.05 ^b^	0.62 ± 0.02 ^a^	4.57 ± 0.12
C	8.87 ± 0.09 ^ab^	93.33 ± 1.35	10.74 ± 0.14 ^b^	1.32 ± 0.00	1155.74 ± 13.77 ^ab^	5.05 ± 0.02 ^ab^	0.63 ± 0.02 ^a^	4.05 ± 0.08
D	9.07 ± 0.19 ^b^	93.33 ± 1.92	10.85 ± 0.10 ^b^	1.30 ± 0.09	1181.95 ± 27.59 ^b^	5.09 ± 0.04 ^b^	0.59 ± 0.01 ^a^	4.24 ± 0.19
E	8.48 ± 0.19 ^ab^	93.33 ± 1.35	10.52 ± 0.10 ^ab^	1.36 ± 0.07	1102.58 ± 27.01 ^ab^	4.97 ± 0.04 ^ab^	0.59 ± 0.01 ^a^	4.41 ± 0.39

Abbreviations are as follows: FBW, final body weight; SR, survival rate; FI, feed intake; FCR, feed conversion ratio; WGR, weight gain rate; SGR, specific growth rate; CF, condition factor; HIS, hepatopancreas somatic indices. Values are presented as mean ± SEM (n = 4). The values in the same row with different superscripts are significantly different (*p* < 0.05).

**Table 4 animals-12-02550-t004:** Effects of compound feed attractants on whole body and muscle composition of the *Litopenaeus vannamei* (wet basis).

Groups	Moisture (%)	Ash (%)	Crude Lipid (%)	Crude Protein (%)
Body				
P	77.35 ± 0.70 ^ab^	3.34 ± 0.09	1.70 ± 0.26 ^a^	16.37 ± 0.35 ^ab^
N	76.53 ± 0.27 ^a^	3.28 ± 0.07	1.64 ± 0.08 ^a^	17.00 ± 0.19 ^b^
A	77.08 ± 0.45 ^ab^	3.18 ± 0.10	1.87 ± 0.13 ^a^	16.71 ± 0.24 ^ab^
B	75.70 ± 0.24 ^a^	3.21 ± 0.15	2.37 ± 0.13 ^b^	17.07 ± 0.22 ^b^
C	76.64 ± 1.18 ^a^	3.21 ± 0.15	1.84 ± 0.17 ^a^	17.01 ± 0.97 ^b^
D	77.62 ± 0.66 ^ab^	3.12 ± 0.10	1.72 ± 0.06 ^a^	16.18 ± 0.29 ^ab^
E	79.04 ± 0.44 ^b^	3.03 ± 0.10	1.41 ± 0.19 ^a^	15.44 ± 0.24 ^a^
Muscle				
P	73.86 ± 0.81	0.57 ± 0.06	1.08 ± 0.10	22.02 ± 0.41
N	74.51 ± 0.82	0.55 ± 0.02	0.93 ± 0.17	21.83 ± 0.33
A	74.93 ± 0.38	0.61 ± 0.03	0.81 ± 0.09	21.88 ± 0.30
B	74.41 ± 0.29	0.52 ± 0.04	0.93 ± 0.06	22.13 ± 0.13
C	74.52 ± 0.65	0.60 ± 0.03	0.85 ± 0.09	22.26 ± 0.37
D	74.59 ± 0.18	0.49 ± 0.07	0.87 ± 0.04	22.23 ± 0.05
E	74.60 ± 0.18	0.74 ± 0.20	0.83 ± 0.03	22.45 ± 0.13

Note: Effects of five compound feed attractants on whole body and muscle composition related parameters of white shrimp. Data are presented as the mean ± SEM (n = 4). Mean values with unlike letters are significantly different (*p* < 0.05).

**Table 5 animals-12-02550-t005:** Effects of compound feed attractants on morphological measurements in midgut of the *Litopenaeus vannamei*.

Groups	FH	FW	MT
P	43.55 + 5.51	50.56 + 9.02	66.86 + 4.32 ^ab^
N	33.89 + 5.34	39.70 + 2.35	62.54 + 7.83 ^ab^
A	37.60 + 6.88	37.33 + 6.52	65.33 + 3.59 ^ab^
B	43.17 + 2.28	37.97 + 6.71	60.72 + 3.24 ^ab^
C	50.52 + 11.94	47.21 + 5.43	74.21 + 6.31 ^b^
D	36.44 + 4.17	38.01 + 2.62	58.56 + 2.20 ^ab^
E	36.03 + 5.60	32.33 + 5.74	56.63 + 3.75 ^a^

Note: MT, FH, and FW represent intestinal muscle thickness, fold height, and fold width, respectively. Values are presented as mean ± SEM (n = 4). The values in the same row with different superscripts are significantly different (*p* < 0.05).

## Data Availability

The data that support the findings of this study are available on request from the corresponding author. The data are not publicly available due to ethical reason, data will be available upon request from the corresponding author.

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
