# Peer review of "Effects of Compound Feed Attractants on Growth Performance, Feed Utilization, Intestinal Histology, Protein Synthesis, and Immune Response of White Shrimp (Litopenaeus Vannamei)"

_animals, 2022, doi:10.3390/ani12192550_

Round 1

Reviewer 1 Report

This manuscript has certain practical significance for aquaculture of shrimp. Growth, intestinal health, protein synthesis and innate immune were analyzed in L. vannamei for evaluate the nutritional effects of different attractants. There are several questions need to be answer and revise.

1. In the section of introduction, these five attractants were introduced detailed. However, more than this, authors should focus on explaining why such complex attractants need to supplemented in mature practical diet for L. vannamei. After all, the use of complex attractants in the diet can not solve the problem of innate slow feed intake speed of L. vannamei proposed by authors.

2. In table 1, there are still different content of crude protein among different experimental diets, with the highest difference even exceeding 3%. How to explain that this discrepancy did not affect the experimental results?

3. In the section of sampling and chemical analysis, the experimental equipment used should be marked with the manufacturer, model and other information in parentheses.

4. In line 138 to 145, the methods and kit code for testing these parameters should be listed.

5. Please check these formulas carefully. (Line 173 to 183) Something wrong were observed among them. They are confused. Also, the “calculating factor” should be changed to “condition factor”, and the “hepatopancreas somatic indices” should be changed to “hepatosomatic index”.

6. Please check the table 4. Whether the data on body and muscle lipid contents is backwards. Why is it higher in muscle than in the body? Also, whether the data is based on wet basis or dry basis should be stated.

7. Figure 4-F is too blurry. Please confirm the resolution of this part. Or make it a single figure or a table.

8. In figure 5, 6, and 7, the spacings should be widened among all experimental group to make a clearer view for readers.

9. In figure 4, we can observe that the intestinal histology of shrimp in B and D groups were significantly different form that in P group, especially the intestinal fold. Why the authors did not discuss this result? Addition, “intestinal histology” is a more accurate description than “intestinal morphology” both in the text and the title.

10. In the section of discussion, authors should reduce the introduction of the meaning od parameters and instead make a more focused and meaningful discussion.

Author Response

Comments and Suggestions for Authors:
-Reviewer 1

This manuscript has certain practical significance for aquaculture of shrimp. Growth, intestinal health, protein synthesis and innate immune were analyzed in L. vannamei for evaluate the nutritional effects of different attractants. There are several questions need to be answer and revise.

  1. In the section of introduction, these five attractants were introduced detailed. However, more than this, authors should focus on explaining why such complex attractants need to supplemented in mature practical diet for  vannamei. After all, the use of complex attractants in the diet can not solve the problem of innate slow feed intake speed of L. vannameiproposed by authors.

Response: Thanks for your valuable suggestion, we have addede it. Complex attractants used in this study was to overcome the disadvantage of single attractant, and the complex attractants were also commonly used in the commercial feed. Besides, through the regression analysis, we can also find the better attranctants among five of them.

  1. In table 1, there are still different content of crude protein among different experimental diets, with the highest difference even exceeding 3%. How to explain that this discrepancy did not affect the experimental results?

Response: In this study, for example, shrimp in groups B and D obtained the best growth performance, while the crude protein of B and D diets were 45.57% and 43.32%, respectively. Overall, based on the growth performance and other results, we thought that the crude protein content in diet was not correlated with the growth performance of shrimp.

  1. In the section of sampling and chemical analysis, the experimental equipment used should be marked with the manufacturer, model and other information in parentheses.

Response: Thanks for your valuable suggestion, we have added them.

  1. In line 138 to 145, the methods and kit code for testing these parameters should be listed.

Response: Thanks for your valuable suggestion, we have listed them, and the kit code were added after these parameters.

  1. Please check these formulas carefully. (Line 173 to 183) Something wrong were observed among them. They are confused. Also, the “calculating factor” should be changed to “condition factor”, and the “hepatopancreas somatic indices” should be changed to “hepatosomatic index”.

Response: Thanks for your valuable suggestion, we have corrected them.

  1. Please check the table 4. Whether the data on body and muscle lipid contents is backwards. Why is it higher in muscle than in the body? Also, whether the data is based on wet basis or dry basis should be stated.

Response: Sorry for this mistake, we have converted the crude lipid of muscle from the dry weight into the wet weight.

  1. Figure 4-F is too blurry. Please confirm the resolution of this part. Or make it a single figure or a table.

Response: Thanks for your valuable suggestion, we have revised it as a table.

  1. In figure 5, 6, and 7, the spacings should be widened among all experimental group to make a clearer view for readers.

Response: Thanks for your valuable suggestion, we have revised them.

  1. In figure 4, we can observe that the intestinal histology of shrimp in B and D groups were significantly different form that in P group, especially the intestinal fold. Why the authors did not discuss this result? Addition, “intestinal histology” is a more accurate description than “intestinal morphology” both in the text and the title.

Response: Thanks for your valuable suggestion, we have added them.

  1. In the section of discussion, authors should reduce the introduction of the meaning od parameters and instead make a more focused and meaningful discussion.

Response: Thanks for your valuable suggestion, we have revised them.

Reviewer 2 Report

This study evaluated the effect of compound feed attractants on growth performance, feed utilization, intestinal morphology, protein synthesis, and immune response of L. vannamei. Overall, the experiment design was reasonable, and the conclusion was supported by solid results. However, the manuscript need to be revised before it could be accepted.

Major points:

1.      There are lots of grammatical errors, and some sentences are imprecise and verbose, the authors are recommended to check the entire manuscript to improve the language.

2.      Tables and figures: One should be able to understand these without consulting the text and the legends should describe the design, including n, the statistics, and abbreviations.

3.      The Conclusion needs some revision and condense, the authors should focus on the main results of this study.

4.      Please check the format of the references, and some references are incomplete.

5.      Why did this experiment was only lasted for 7 weeks, not 8 weeks?

6.      Have you evaluated the stability of the reference gene elongation factor 1α, because it is important to the gene expression data.

Minor points:

Line43: Please add “the” after “fed”.

Line67: Please replace “could” to “can”.

Line82: Please replace “,” between “feed intake” and “ feed utilization”.

Line91: Please add “.” after “)”.

Lines 113-114: How was this done?

Line134: Please change “high-density lipoprotein” with “high density lipoprotein cholesterin”.

Line134: Please change “low-density lipoprotein” with “low density lipoprotein cholesterin”.

Line151: Please add the manufacturer message about “Green Pro Taq HS Premix II”.

Line156: What is specific software about “PrimerQuest tool”?

Line194: Please add “the” between “all” and “experimental”.

Line203: Please add “the” between “all” and “experimental”.

Line208: Please delete “among all experimental groups” between “shrimp” and “showed”.

Line312: Please add “the” between “all” and “experimental”.

Line330: Please deleted “significant”.

Line331: Please change “enhancing” with “enhance”.

Line332: Please change “enhancing” with “enhance”.

Line404: Please change “and” between “feed intake” and “growth performance” with “,”.

Figure 2: Please change “capacit” with “capacity”.

Figure 4: Please change “amongt” with “among”.

Author Response

Comments and Suggestions for Authors:

-Reviewer 2

This study evaluated the effect of compound feed attractants on growth performance, feed utilization, intestinal morphology, protein synthesis, and immune response of L. vannamei. Overall, the experiment design was reasonable, and the conclusion was supported by solid results. However, the manuscript need to be revised before it could be accepted.

Major points:

  1. There are lots of grammatical errors, and some sentences are imprecise and verbose, the authors are recommended to check the entire manuscript to improve the language.

Response: Thanks for your valuable suggestion, we have revised them.

  1. Tables and figures: One should be able to understand these without consulting the text and the legends should describe the design, including n, the statistics, and abbreviations.

Response: Thanks for your valuable suggestion, we have corrected it.

  1. The Conclusion needs some revision and condense, the authors should focus on the main results of this study.

Response: Thanks for your valuable suggestion, we have rewvised it.

  1. Please check the format of the references, and some references are incomplete.

Response: Thanks for your valuable suggestion, we have added them.

  1. Why did this experiment was only lasted for 7 weeks, not 8 weeks?

Response: Because of the stormy weather and sudden drop of temperature in the late period of the feeding experiment, it is not conducive to the continuation of the feeding experiment.

  1. Have you evaluated the stability of the reference gene elongation factor 1α, because it is important to the gene expression data.

Response: Yes, we evaluated the stability of the reference gene elongation factor 1α through pre-experiments over different time periods.

Minor points:

Line43: Please add “the” after “fed”.

Response: Thanks for your valuable suggestion, we have corrected it.

Line67: Please replace “could” to “can”.

Response: Thanks for your valuable suggestion, we have corrected it.

Line82: Please replace “,” between “feed intake” and “ feed utilization”.

Response: Thanks for your valuable suggestion, we have corrected it.

Line91: Please add “.” after “)”.

Response: Thanks for your valuable suggestion, we have added it.

Lines 113-114: How was this done?

Response: Thanks for your valuable suggestion.The diurnal cycle was controlled by the light, the water quality was maintained through the exchange of water (three times in one week)

Line134: Please change “high-density lipoprotein” with “high density lipoprotein cholesterin”.

Response: Thanks for your valuable suggestion, we have corrected it.

Line134: Please change “low-density lipoprotein” with “low density lipoprotein cholesterin”.

Response: Thanks for your valuable suggestion, we have corrected it.

Line151: Please add the manufacturer message about “Green Pro Taq HS Premix II”.

Response: Thanks for your valuable suggestion, we have added it.

Line156: What is specific software about “PrimerQuest tool”?

Response: Thanks for your valuable suggestion, we have added it.

Line194: Please add “the” between “all” and “experimental”.

Response: Thanks for your valuable suggestion, we have added it.

Line203: Please add “the” between “all” and “experimental”.

Response: Thanks for your valuable suggestion, we have added it.

Line208: Please delete “among all experimental groups” between “shrimp” and “showed”.

Response: Thanks for your valuable suggestion, we have deleted it.

Line312: Please add “the” between “all” and “experimental”.

Response: Thanks for your valuable suggestion, we have added it.

Line330: Please deleted “significant”.

Response: Thanks for your valuable suggestion, we have deleted it.

Line331: Please change “enhancing” with “enhance”.

Response: Thanks for your valuable suggestion, we have corrected it.

Line332: Please change “enhancing” with “enhance”.

Response: Thanks for your valuable suggestion, we have corrected it.

Line404: Please change “and” between “feed intake” and “growth performance” with “,”.

Response: Thanks for your valuable suggestion, we have deleted it.

Figure 2: Please change “capacit” with “capacity”.

Response: Thanks for your valuable suggestion, we have corrected it.

Figure 4: Please change “amongt” with “among”.

Response: Thanks for your valuable suggestion, we have corrected it.

Reviewer 3 Report

This is an interesting study, but the methodology is not clear to me and should probably more detailed, specially regarding the choice of the diets’ composition (see below).

Check the way references are presented as they are in the text as “author´s name, date) but the final list of references is both presented in alphabetic order and with numbers. Please check the “Instruction for authors” of the journal.

You will find below some suggestions for small corrections and some issues that may deserve some reflection.

In the beginning of the introduction (“L. vannamei is one of the most widely cultured shrimp species worldwide due to its 37 good taste, rapid growth, and excellent salinity adaptability (Chen, et al., 2018). In 2018, 38 the global production of Pacific white shrimp exceed 4.9 million tons…”) the name of the species should be presented in full, as it is the first time that it is mentioned. Additionally, the current designation (Pacific white shrimp) should be properly associated to the Latin name to be clear that you are referring to the same species.

Line 39: “…Pacific white shrimp exceed 4.9 million…” should be “…Pacific white shrimp exceeded 4.9 million…”

Line 48: “So the attractants in the feed…” should be “…So, the attractants in the feed…”

Lines 55-66: in this part of the introduction, the information seems a little bit confused. Is presents information that we expect to find in the Methods section (the feed attractants used in the study) and it is not consistent in terms of the information provided (origin, main nutrients in their composition, specific characteristics, as the strong odor, and how they are produced). In the following text, from the same paragraph, results already obtained in other studies are presented, but, once again, talking about the various compounds in a confusing way.

I suggest that you present the information related to each of the compounds in a more systematic way, thus taking the opportunity to base the choice of each one of them for this study.

Lines 83-84: “This study aimed to find out the suitable attractant combinations for L. vannamei.” The objective, presented in this way, seems both very vague and very ambitious, as if the results were intended to be a definitive solution to the issue under analysis. I suppose there are at least other compounds and even other combinations of the compounds used that might be good solutions to this question as well. I suggest that you rewrite the objective to better represent what you have done.

Table 1: The caption could be more complete as the table does not present only the formulation of the diets. Also, the format of the numbers should be more homogeneous (all with 2 decimal places).

Regarding its contents, is there any reason why diet “N” has 15% soybean meal, while all the others have only 7%? I would also like you to explain better why you decided to prepare diets with 4 of the 5 compounds studied. How does this option allow you to conclude about the effect of a particular attractant?

The values presented, related with the proximate analysis, are the result of a single analysis or did you analyze replicates? Is so, how many? And, in this case, the mean value should be presented with the respective standard deviation. Are there significant differences between the diets is relation to any of the parameters?

Lines 110-111: Did all the shrimp weigh exactly 0.17 g?

Lines 125-127: Why did you use only 3 shrimp to determine weight and body length and not record these values ​​for all shrimp that were taken from the tanks for the different analyses? In this way you would have a more statistically significant value.

Lines 204-206: I may be missing something, but it seems to me that it would be important to explain better how you can conclude about the effect of a particular attractant when you always have a mixture of 4 in each diet.

Lines 225-226: “Effects of five compound feed attractants on whole-body and muscle composition related parameters of white shrimp. Moisture, ash, crude lipid, crude protein.” Why is this information included in the footnote of table 4?

Figure 1 (legend): “biochemical paraameters”. Correct it please.

Line 231: “…and LDC-C…” Do you mean LDL-C?

Figure 4, line 282: “In figure 4 (A, F), arrows MT, FH, and FW …” do you mean (A, E)? And the arrows, MT, FH, and FW you mention are only present in photograph A. They should be included in all images and the arrows need to be thicker.  I also suggest that the letter identifying the diet in each image is always place in the same position.

Lines 275-276: “(…figure 4 A, B, C, D, E)…” could be “(…figure 4 A- E)…”

Lines 287-288: “Immune response and protein synthesis related genes expression in hepatopancreas in L. vannamei are presented in figure 5 and figure 6.” should be “… are presented in figure 5 and figure 6, respectively.”

Figure 5: The differences mentioned in the text regarding the expression of toll are not marked in the graphic

Line 329: “…shrimp fed the P, N. A, and…” should be “shrimp fed the P, N, A, and…”

Lines 300-333: Please check the text describing the results regarding the expression of relish as they do not seem to be describing what is presented in the respective graphic (figure 7):

 “The expression level of relish in shrimp fed the B diet was significantly higher than those fed the other diets (P < 0.05), in shrimp fed the B diet was significantly higher than those fed the P, N, A, D, and E diets (P < 0.05).”

Discussion:

How can you conclude that “Squid paste, and shrimp paste contributed more to the WG and FI, while yeast extract and fish soluble may have negative effects” (lines 344-345), because “the shrimp fed the B and D diets obtained better growth performance than those fed the other diets” (lines 343-344 and 372-374)?

Why do you say that “yeast extract and fish soluble may have negative effects” (line 345). Negative how? You also say that “Contrary to our findings, there were previous studies showed that the feed conversion ratio of white shrimp was reduced by consuming a yeast culture feed supplement…” (lines 360-362). Why do you consider it is the opposite if you also refer to negative effects?

Lines 378-379: “In our experiment, T-CHO and TG were no differences among all groups.” I suggest you change to “In our experiment, T-CHO and TG levels present no significant differences among all groups.”

I do not understand the information related to HDL and LDL (lines 379-385) - Are you referring to humans or to the shrimps? Do shrimps have a problem with atherosclerosis? And then, in the remaining text of the paragraph, can we really conclude about the ability that shrimps have to remove and/or transport cholesterol when fed with a specific attractant, based on the obtained results?

How do you conclude that “…indicating that the shrimp in N and A groups had the strongest ability to transport cholesterol into the cells, while D group was the weakest.” Based on the values determined for HDL-C levels? HDL are involved in reverse transport, as you mentioned, so how do we conclude about the transport into the cells?

Lines 395-398: To what kind of damage are you referring?

Lines 403-409: You present again the results obtained regarding the expression of sod, but you don’t discuss them. What do they mean or suggest? The same happens in relation to MDA, T-AOC and NOS. A deeper discussion should be presented.

Line 428: “…against pathogens literature…”. Literature? Please correct.

Conclusions would be easier to understand if the different diets mentioned are presented also by the letters that identify them throughout the article. Also, what is the better diet, considering the aim of the study presented in the beginning of the article? What is the better choice for shrimp producers? Do you think you have enough information?

Author Response

Comments and Suggestions for Authors:

-Reviewer 3

This is an interesting study, but the methodology is not clear to me and should probably more detailed, specially regarding the choice of the diets’ composition (see below).

Check the way references are presented as they are in the text as “author´s name, date) but the final list of references is both presented in alphabetic order and with numbers. Please check the “Instruction for authors” of the journal.

Response: Thanks for your valuable suggestion, we have corrected them.

You will find below some suggestions for small corrections and some issues that may deserve some reflection.

In the beginning of the introduction (“L. vannamei is one of the most widely cultured shrimp species worldwide due to its 37 good taste, rapid growth, and excellent salinity adaptability (Chen, et al., 2018). In 2018, 38 the global production of Pacific white shrimp exceed 4.9 million tons…”) the name of the species should be presented in full, as it is the first time that it is mentioned. Additionally, the current designation (Pacific white shrimp) should be properly associated to the Latin name to be clear that you are referring to the same species.

Response: Thanks for your reminder, we have revised it.

Line 39: “…Pacific white shrimp exceed 4.9 million…” should be “…Pacific white shrimp exceeded 4.9 million…”

Response: Thanks for your valuable suggestion, we have corrected it.

Line 48: “So the attractants in the feed…” should be “…So, the attractants in the feed…”

Response: Thanks for your valuable suggestion, we have corrected it.

Lines 55-66: in this part of the introduction, the information seems a little bit confused. Is presents information that we expect to find in the Methods section (the feed attractants used in the study) and it is not consistent in terms of the information provided (origin, main nutrients in their composition, specific characteristics, as the strong odor, and how they are produced). In the following text, from the same paragraph, results already obtained in other studies are presented, but, once again, talking about the various compounds in a confusing way.

I suggest that you present the information related to each of the compounds in a more systematic way, thus taking the opportunity to base the choice of each one of them for this study.

Response: Thanks for your valuable suggestion, we have revised them.

Lines 83-84: “This study aimed to find out the suitable attractant combinations for L. vannamei.” The objective, presented in this way, seems both very vague and very ambitious, as if the results were intended to be a definitive solution to the issue under analysis. I suppose there are at least other compounds and even other combinations of the compounds used that might be good solutions to this question as well. I suggest that you rewrite the objective to better represent what you have done.

Response: Thanks for your valuable suggestion, we have revised them.

Table 1: The caption could be more complete as the table does not present only the formulation of the diets. Also, the format of the numbers should be more homogeneous (all with 2 decimal places).

Response: Thanks for your valuable suggestion, we have added it.

Regarding its contents, is there any reason why diet “N” has 15% soybean meal, while all the others have only 7%? I would also like you to explain better why you decided to prepare diets with 4 of the 5 compounds studied. How does this option allow you to conclude about the effect of a particular attractant?

Response: Thanks for your valuable suggestion. In the commercial feed (P), the total proportion of yeast extract, fish soluble pulp and squid visceral powder was 8%, and the attractant combination as a whole was a single variable, so the total proportion of other attractant combinations in the feed group was 8%, and the content of soybean meal in the N group was 15%, which was equivalent to 8% soybean meal replacing 8% attractant combination.

The values presented, related with the proximate analysis, are the result of a single analysis or did you analyze replicates? Is so, how many? And, in this case, the mean value should be presented with the respective standard deviation. Are there significant differences between the diets is relation to any of the parameters?

Response: Thanks for your valuable suggestion. The results of four replicates were presented as mean with SEM, all data were subjected to one-way analysis of variance (ANOVA) followed by Duncan Range Test to determine significant differences among treatments using SPSS 21 (SPSS, Chicago, IL, USA). There are no significant differences between the diets is relation to any of the parameters.

Lines 110-111: Did all the shrimp weigh exactly 0.17 g?

Response: Thanks for your valuable suggestion. We calculated the initial weight of (0.71±0.003) g by selecting individual and 30 healthy and uniform shrimp and adding the total weight, instead of weighing each shrimp, here “±0.003g” is omitted to keep the decimal points aligned.

Lines 125-127: Why did you use only 3 shrimp to determine weight and body length and not record these values for all shrimp that were taken from the tanks for the different analyses? In this way you would have a more statistically significant value.

Response: Thanks for your valuable suggestion. In our previous studies, this sampling method is usually used to carry out morphological parameters.

Lines 204-206: I may be missing something, but it seems to me that it would be important to explain better how you can conclude about the effect of a particular attractant when you always have a mixture of 4 in each diet.

Response: Thanks for your valuable suggestion. We have mentioned it in Lines 233-235.

Lines 225-226: “Effects of five compound feed attractants on whole-body and muscle composition related parameters of white shrimp. Moisture, ash, crude lipid, crude protein.” Why is this information included in the footnote of table 4?

Response: Thanks for your valuable suggestion, we have revised them.

Figure 1 (legend): “biochemical paraameters”. Correct it please.

Response: Thanks for your valuable suggestion, we have corrected it.

Line 231: “…and LDC-C…” Do you mean LDL-C?

Response: Thanks for your valuable suggestion, we have corrected it.

Figure 4, line 282: “In figure 4 (A, F), arrows MT, FH, and FW …” do you mean (A, E)? And the arrows, MT, FH, and FW you mention are only present in photograph A. They should be included in all images and the arrows need to be thicker.  I also suggest that the letter identifying the diet in each image is always place in the same position.

Response: Thanks for your valuable suggestion, we have revised it.

Lines 275-276: “(…figure 4 A, B, C, D, E)…” could be “(…figure 4 A- E)…”

Response: Thanks for your suggestion, we have revised it.

Lines 287-288: “Immune response and protein synthesis related genes expression in hepatopancreas in L. vannamei are presented in figure 5 and figure 6.” should be “… are presented in figure 5 and figure 6, respectively.”

Response: Thanks for your valuable suggestion, we have revised it.

Figure 5: The differences mentioned in the text regarding the expression of toll are not marked in the graphic

Response: Thanks for your suggestion, we have marked it in the graphic (Figure 5).

Line 329: “…shrimp fed the P, N. A, and…” should be “shrimp fed the P, N, A, and…”

Response: Thanks for your valuable suggestion, we have revised it.

Lines 300-333: Please check the text describing the results regarding the expression of relish as they do not seem to be describing what is presented in the respective graphic (figure 7):

 “The expression level of relish in shrimp fed the B diet was significantly higher than those fed the other diets (P < 0.05), in shrimp fed the B diet was significantly higher than those fed the P, N, A, D, and E diets (P < 0.05).”

Response: Thanks for your valuable suggestion. We have presented the result of relish as  “The expression level of relish in shrimp fed the B diet was significantly higher than those fed the other diets (P < 0.05), in shrimp fed the B diet was significantly higher than those fed the P, N, A, D, and E diets (P < 0.05).” in figure 7.

Discussion:

How can you conclude that “Squid paste, and shrimp paste contributed more to the WG and FI, while yeast extract and fish soluble may have negative effects” (lines 344-345), because “the shrimp fed the B and D diets obtained better growth performance than those fed the other diets” (lines 343-344 and 372-374)?

Response: Thanks for your valuable suggestion. We concluded that “Squid paste, and shrimp paste contributed more to the WG and FI, while yeast extract and fish soluble may have negative effects” by Regression analysis. In regression analysis, single attractant in each diet (A- E) was used as independent variable, WG and FI were used as dependent variables.

Why do you say that “yeast extract and fish soluble may have negative effects” (line 345). Negative how? You also say that “Contrary to our findings, there were previous studies showed that the feed conversion ratio of white shrimp was reduced by consuming a yeast culture feed supplement…” (lines 360-362). Why do you consider it is the opposite if you also refer to negative effects?

Response: Thanks for your valuable suggestion. We concluded that “Squid paste, and shrimp paste contributed more to the WG and FI, while yeast extract and fish soluble may have negative effects.” (line 345-346) by regression analysis (line 187-190), We consider “Contrary to our findings, there were previous studies showed that the feed conversion ratio of white shrimp was reduced by consuming a yeast culture feed supplement…” (lines 360-362), because yeast extract did not improve feed utilization in this study.

Lines 378-379: “In our experiment, T-CHO and TG were no differences among all groups.” I suggest you change to “In our experiment, T-CHO and TG levels present no significant differences among all groups.”

Response: Thanks for your valuable suggestion, we have revised it.

I do not understand the information related to HDL and LDL (lines 379-385) - Are you referring to humans or to the shrimps? Do shrimps have a problem with atherosclerosis? And then, in the remaining text of the paragraph, can we really conclude about the ability that shrimps have to remove and/or transport cholesterol when fed with a specific attractant, based on the obtained results?

Response: Thanks for your valuable suggestion, we have revised it.

How do you conclude that “…indicating that the shrimp in N and A groups had the strongest ability to transport cholesterol into the cells, while D group was the weakest.” Based on the values determined for HDL-C levels? HDL are involved in reverse transport, as you mentioned, so how do we conclude about the transport into the cells?

Response: Thanks for your valuable suggestion, we have revised it.

Lines 395-398: To what kind of damage are you referring?

Response: Thanks for your valuable suggestion, we have added it.

Lines 403-409: You present again the results obtained regarding the expression of sod, but you don’t discuss them. What do they mean or suggest? The same happens in relation to MDA, T-AOC and NOS. A deeper discussion should be presented.

Response: Thanks for your valuable suggestion, we have added it.

Line 428: “…against pathogens literature…”. Literature? Please correct.

Response: We are sorry that “literature” was not accurate, we want to say “pathogens”, we have deleted it..

Conclusions would be easier to understand if the different diets mentioned are presented also by the letters that identify them throughout the article. Also, what is the better diet, considering the aim of the study presented in the beginning of the article? What is the better choice for shrimp producers? Do you think you have enough information?

Response: Thanks for your valuable suggestion, we have revised it.

Round 2

Reviewer 1 Report

This manuscript has certain practical significance for aquaculture of shrimp. Growth, intestinal health, protein synthesis and innate immune were analyzed in L. vannamei for evaluate the nutritional effects of different attractants. There are several questions need to be answer and revise.

1. In the section of introduction, these five attractants were introduced detailed. However, more than this, authors should focus on explaining why such complex attractants need to supplemented in mature practical diet for L. vannamei. After all, the use of complex attractants in the diet can not solve the problem of innate slow feed intake speed of L. vannamei proposed by authors.

2. In table 1, there are still different content of crude protein among different experimental diets, with the highest difference even exceeding 3%. How to explain that this discrepancy did not affect the experimental results?

3. In the section of sampling and chemical analysis, the experimental equipment used should be marked with the manufacturer, model and other information in parentheses.

4. In line 138 to 145, the methods and kit code for testing these parameters should be listed.

5. Please check these formulas carefully. (Line 173 to 183) Something wrong were observed among them. They are confused. Also, the “calculating factor” should be changed to “condition factor”, and the “hepatopancreas somatic indices” should be changed to “hepatosomatic index”.

6. Please check the table 4. Whether the data on body and muscle lipid contents is backwards. Why is it higher in muscle than in the body? Also, whether the data is based on wet basis or dry basis should be stated.

7. Figure 4-F is too blurry. Please confirm the resolution of this part. Or make it a single figure or a table.

8. In figure 5, 6, and 7, the spacings should be widened among all experimental group to make a clearer view for readers.

9. In figure 4, we can observe that the intestinal histology of shrimp in B and D groups were significantly different form that in P group, especially the intestinal fold. Why the authors did not discuss this result? Addition, “intestinal histology” is a more accurate description than “intestinal morphology” both in the text and the title.

10. In the section of discussion, authors should reduce the introduction of the meaning od parameters and instead make a more focused and meaningful discussion.

Author Response

(The authors gave the same response as above.)

Reviewer 3 Report

I acknowledge and appreciate the efforts of the authors to carry out the suggested changes. It seems to me that there are still some minor issues that can be checked.

1) Line 20: “…intestinal morphology…” should it be histology?

2) Regarding the proximate analysis, I was also referring to the diets (table 1) where the proximate composition of the diets is presented without SEM.

Lines 130-131: “840 healthy and uniform-sized shrimp (initial body weight of 0.71 g)…”; I don´t understand when you say that “… here “±0.003g” is omitted to keep the decimal points aligned”. What do you mean by “aligned”? In my opinion, this value should be presented as it is more important than the alignment of the text, which I believe can be properly formatted.

3) Lines 204-206: I may be missing something, but it seems to me that it would be important to explain better how you can conclude about the effect of a particular attractant when you always have a mixture of 4 in each diet.

Response: Thanks for your valuable suggestion. We have mentioned it in Lines 233 235.

I was not able to find this information in these mentioned lines.

4) When you say “We concluded that “Squid paste, and shrimp paste contributed more to the WG and FI, while yeast extract and fish soluble may have negative effects” by Regression analysis. In regression analysis, single attractant in each diet (A E) was used as independent variable, WG and FI were used as dependent variables.”

You could probably consider presenting and explaining better the results obtained when you carried out the regression analysis.

Author Response

Comments and Suggestions for Authors

-Reviewer 3

I acknowledge and appreciate the efforts of the authors to carry out the suggested changes. It seems to me that there are still some minor issues that can be checked.

  • Line 20: “…intestinal morphology…” should it be histology?

Response: Thanks for your valuable suggestion, we have revised it.

  • Regarding the proximate analysis, I was also referring to the diets (table 1) where the proximate composition of the diets is presented without SEM.

Response: Thanks for your valuable suggestion, in most previous studies, the proximate composition of the diets is presented without SEM, and due to the size limitation of table 1, the proximate composition of the diets is presented without SEM. To confirm your review more convenient, we have listed the proximate composition of the diets is presented with SEM. As following:

Groups

Crude protein

Crude lipid

Moisture

Ash

P

43.00±0.00

6.60±0.00

6.78±0.00

16.13±0.00

N

42.07±0.00

5.84±0.00

6.44±0.00

15.97±0.00

A

44.64±0.00

5.77±0.00

6.44±0.00

16.41±0.00

B

45.57±0.00

5.95±0.00

6.88±0.00

16.30±0.00

C

44.37±0.00

5.17±0.00

7.35±0.00

16.36±0.00

D

43.32±0.00

6.16±0.00

6.48±0.00

15.56±0.00

E

45.11±0.00

6.14±0.00

8.19±0.00

16.13±0.00

Lines 130-131: “840 healthy and uniform-sized shrimp (initial body weight of 0.71 g)…”; I don´t understand when you say that “… here “±0.003g” is omitted to keep the decimal points aligned”. What do you mean by “aligned”? In my opinion, this value should be presented as it is more important than the alignment of the text, which I believe can be properly formatted.

Response: Thanks for your valuable suggestion, we would like to say that “0.71 g” is the initial weight of shrimp, which was obtained by statistical analysis is (0.710±0.003) g. We have revised it as “(0.71±0.00) g”.

3) Lines 204-206: I may be missing something, but it seems to me that it would be important to explain better how you can conclude about the effect of a particular attractant when you always have a mixture of 4 in each diet.

Response: Thanks for your valuable suggestion, it is a kind of multiple regression analysis, attractants in each diet (A-E) was used as independent variable, WG and FI were used as dependent variables. Regression equations were obtained between the five feed attractants and FI, and WG by multiple regression analysis, respectively.

4) When you say “We concluded that “Squid paste, and shrimp paste contributed more to the WG and FI, while yeast extract and fish soluble may have negative effects” by Regression analysis. In regression analysis, single attractant in each diet (A E) was used as independent variable, WG and FI were used as dependent variables.”

You could probably consider presenting and explaining better the results obtained when you carried out the regression analysis.

Response: Thanks for your valuable suggestion, we have revised them in the Results and Conclusion.
